# Effect of Annealing Temperature on the Interfacial Microstructure and Bonding Strength of Cu/Al Clad Sheets with a Stainless Steel Interlayer

**DOI:** 10.3390/ma15062119

**Published:** 2022-03-13

**Authors:** Haitao Gao, Hao Gu, Sai Wang, Yanni Xuan, Hailiang Yu

**Affiliations:** 1State Key Laboratory of High Performance Complex Manufacturing, Light Alloys Research Institute, College of Mechanical and Electrical Engineering, Central South University, Changsha 410083, China; gaohaitao@csu.edu.cn (H.G.); guhao0927@csu.edu.cn (H.G.); yuhailiang@csu.edu.cn (H.Y.); 2State Key Laboratory of Rolling and Automation, Northeastern University, Shenyang 110819, China; 3Department of Energy and Power Engineering, School of Energy and Power Engineering, Changsha University of Science and Technology, Changsha 410114, China; xuanyanni@csut.edu.cn

**Keywords:** Cu/Al clad sheet, interlayer, annealing temperature, interfacial reaction, bonding strength

## Abstract

To explore the influence of annealing temperatures on the interfacial structure and peeling strength of Cu/Al clad sheets with a 304 stainless steel foil interlayer, an intermediate annealing treatment was performed at temperatures of 450 °C, 550 °C, and 600 °C, separately. The experimental results indicate that the interfacial atomic diffusion is significantly enhanced by increasing the intermediate annealing temperature. The average peeling strength of the clad sheets annealed at 550 °C can reach 34.3 N/mm and the crack propagation is along the steel/Cu interface, Cu-Al intermetallic compounds layer, and Al matrix. However, after high-temperature annealing treatment (600 °C), the liquid phase is formed at the bonding interface and the clear Cu/steel/Al interface is replaced by the chaotic composite interfaces. The clad sheet broke completely in the unduly thick intermetallic compounds layer, resulting in a sharp decrease in the interfacial bonding strength.

## 1. Introduction

Cu/Al clad sheets have been widely applied in many fields, such as power electronics, aerospace, and electronic communication [1]. The common methods to produce Cu/Al clad sheets are rolling bonding [2], twin-roll casting bonding [3], explosive bonding [4], and diffusion bonding/TLP bonding/diffusion brazing [5]. Different from these traditional techniques, the powder-in-tube method exhibits dominant advantages to fabricate metallic clad sheets with high interfacial bonding strength, which can easily achieve the closure of local defects around the bonding interface and the regulation in thickness and structure of intermetallic compounds (IMCs) [6]. It is a promising and environmentally friendly method to fabricate metallic clad sheets.

The annealing treatment has a significant influence on the interfacial structure and mechanical performance of metallic clad sheets [7,8]. Cu-Al IMCs are easily formed at the bonding interface due to the strong chemical affinity between Cu and Al [9]. As such, a considerable amount of research has been carried out to regulate the structure of Cu-Al IMCs. In the annealing process, Gao et al. [10] found that the CuAl_2_ phase was in possession of the priority formation rather than other Cu-Al IMCs, whose formation sequence was in the order of CuAl_2_, Cu_9_Al_4_, CuAl, and Cu_4_Al_3_ (or Cu_3_Al_2_) [2]. The mathematical relationship between the IMCs layer thickness, annealing time, and annealing temperature was built by Pelzer et al. [11]. Similarly, Lee et al. [12] indicated that the growth of Cu-Al IMCs could be controlled by a diffusion mechanism. Li et al. [13] showed that a thicker IMCs layer induced by the high annealing temperature usually led to a decrease in interfacial bonding strength. Mao et al. [14] indicated that the thickness of the IMCs layer in Cu/Al clad sheet could be controlled within 550 nm after annealing at 250 °C and the peeling strength could reach 39 N/mm. Therefore, effectively controlling the thickness and structure of IMCs is a critical factor to obtaining a high-performance Cu/Al clad sheet.

Our previous research found that the introduction of a 304 stainless steel (SUS304) interlayer could significantly improve the interfacial strength of the Cu/Al clad sheet by optimizing the type and structure of IMCs [15]. In this paper, Cu/Al clad sheets with a SUS304 interlayer were prepared by the powder-in-tube method. The influence of annealing temperature on the interfacial structure and bonding strength was investigated. Furthermore, the interfacial strengthening mechanism was systematically discussed.

## 2. Materials and Methods

The initial materials are the commercial pure copper tube (99.9%, Guangfeng Metal Materials Co., Ltd., Dongguan, China) with an outer diameter of 10 mm and a wall thickness of 1 mm and atomized aluminum powder with an average diameter of 10 μm. The cold-rolled SUS304 foils (Guangfeng Metal Materials Co., Ltd., Dongguan, China) with a thickness of 30 μm are chosen as the interlayer materials. The mechanical parameters of the SUS304 foils have been illustrated in our previous research [15]. Before roll cladding, the oxide layer on the inner surface of the Cu tube is removed by the wire brush. The detailed preparation process of the Cu/Al clad sheets with the SUS304 interlayer is illustrated in Figure 1. Firstly, the Cu cube, SUS304 foil, and Al powder (Hunan Jinhao New Material Technology Co., Ltd., Miluo, China) are manually assembled together to form the composite tube billet. Secondly, the Cu/Al clad sheets are rolled to 1.5 mm after multi-pass cold rolling and then annealed at different temperatures of 450 °C, 550 °C, and 600 °C, which are designated as IFR-450, IFR-550, and IFR-600, respectively. Finally, these annealed samples are further rolled to 0.5 mm. 

To observe the microstructure of the bonding interface and peeling surface, scanning electron microscopy (SEM) was performed using an FEI Quanta 250F (FEI Company, Hillsboro, OR, USA) device with an acceleration voltage of 20 kV and equipped with energy dispersive spectrometry (EDS, Oxford Instruments Group, London, UK). The Al-Cu phase diagram is plotted by the software ‘Binary Alloy Phase Diagrams’ (ASM International, Almere, the Netherlands). The crystalline phases on the peeling surface are detected by D8 ADVANCE X-ray diffraction (XRD, ceramic X-ray tube, Brooke company, karlsruhe, Germany), using Cu K_α_ radiation (λ ≈ 1.54 Å), equipped with a Linx one-dimensional array detector (Liwei wisdom international Co., Ltd., HongKong, China). The angle step, time step, and scanning range are 0.01°, 0.1 s, and 10–100°, respectively. T-peel tests on a TH5000 universal testing machine (Xintianhui Electronic Technology Co., Ltd., Yangzhou, China) are adopted to test the interfacial bonding strength of Cu/Al clad sheets under a crosshead speed of 1 mm/min.

## 3. Results and Discussion

### 3.1. Interfacial Microstructure

Prior research indicated that the peeling strength of Cu/Al clad sheets was mainly determined by the bonding strength between the SUS304 interlayer and the Cu/Al matrix [15]. Figure 2 exhibits the interfacial microstructure of IFR-450 and the corresponding EDS mapping results. A flat bonding interface without visible cracks between the SUS304 interlayer and Cu/Al matrix is formed (Figure 2a), and the IMCs are seldom formed due to the lower intermediate annealing temperature [16], which is proved by the EDS mapping results of the Al/SUS304/Cu bonding interface (Figure 2b–d). The position of the SUS304 interlayer is marked by the distribution of the Fe and Cr elements (Figure 2e,f). In the rolling process, the SUS304 fragments are squeezed into the Cu/Al matrix. Under this situation, the interfacial bonding strength is mainly contributed by the mechanical joggles [17].

When the intermediate annealing temperature increases to 550 °C, obvious IMCs with an average thickness of 10 μm are formed at the bonding interface of the SUS304 interlayer and Al matrix [18], as shown in Figure 3a, which is also strongly proved by the corresponding EDS mapping results (Figure 3b). A small increase in the annealing temperature may result in a significant increment in the diffusion coefficient [19] due to their exponential relationship (Figure 3c). In contrast, the chemical compound type formed between the SUS304 interlayer and Cu matrix [20] is the solid solution (Figure 3d). The position of the SUS304 interlayer is marked by the distribution of Fe and Cr elements (Figure 3e,f). For Cu/Al clad sheets without an interlayer, the Cu-Al IMCs are broken into fragments in the rolling process [6]. Nevertheless, the IMCs formed between the SUS304 interlayer and Al matrix can retain their continuity. The existence of an SUS304 interlayer with a weak deformation capacity can significantly inhibit the crush of Al-SUS304 IMCs and enhance the interfacial bonding strength.

The interfacial microstructure of IFR-600 is exhibited in Figure 4. As the intermediate annealing temperature reaches 600 °C, the clear Cu/SUS304/Al interface disappears and is replaced by the chaotic composite interfaces (Figure 4a). According to the Al-Cu phase diagram (Figure 5), the liquid phase will be formed at the bonding interface with the annealing temperature of 600 °C (red line in Figure 5). In this case, the Cu/SUS304/Al interface is destroyed by the disturbing force induced from the formation process of the liquid phase, resulting in the exfoliation of SUS304 fragments from the Cu matrix and being involved in the liquid phases. Due to the relatively high annealing temperature, the diffusional degree of IFR-600 is much larger than those in IFR-450 and IFR-550 (Figure 2 and Figure 3), which is also proven by our prior research [21]. Based on the EDS mapping results (Figure 4b–d), it can be observed that the SUS304 fragments are surrounded by thick Cu-Al IMCs. The position of the SUS304 interlayer is marked by the distribution of Fe and Cr elements (Figure 4e,f). Moreover, the hardness of Cu-Al IMCs is higher than the Cu/Al matrix, leading to the increase in the thickness reduction of the SUS304 interlayer in the further cold rolling process. For IFR-600, the residual thickness of the SUS304 interlayer is just 5.9 μm, around a third of those in IFR-450 and IFR-550.

### 3.2. Elemental Diffusion across the Bonding Interface

The EDS line results across the Cu/SUS304/Al interface are provided in Figure 6. After roll cladding, the thickness reduction in the SUS304 interlayers in IFR-450, IFR-550, and IFR-600 are 11.6 μm, 10.2 μm, and 24.1 μm, respectively. In general, the thickness reduction in the SUS304 interlayers should decrease with the increment of intermediate annealing temperature due to the aggravate softening of the Cu/Al matrix. Nevertheless, the formation of liquid phases at the bonding interface leads to the SUS304 interlayer being surrounded by the Cu-Al IMCs and a large thickness reduction in the SUS304 interlayer (Figure 4). On the other hand, the interfacial atomic diffusion is also enhanced by the increase in annealing temperature, which may also increase the corrosion resistance of the clad sheets [22]. Compared with IFR-450, the width of the diffusional layer for the Al/SUS304 interface and Cu/SUS304 interface in IFR-550 is increased from 2.2 μm to 20.1 μm and from 1.1 μm to 3.5 μm (Figure 6a,b). As for IFR-600, the SUS304 fragments are totally surrounded by the thick Cu-Al IMCs, which is strongly proven by the EDS line results (Figure 6c).

### 3.3. Peeling Surface of the Clad Sheets

The XRD patterns of the peeling surface of IFR samples are provided in Figure 7. It can be seen that the intermediate annealing temperature has a significant impact on the type and content of crystalline phases on the peeling surface. The main crystalline phases in the peeling surface of the Cu side for IFR-450 and IFR-550 are Cu, Al, Cu_9_Al_4,_ and CuAl_2_. The interfacial elemental diffusion is enhanced by the increase in the intermediate annealing temperature, resulting in the increment of Cu-Al IMCs, especially the Cu_9_Al_4_ phase (Figure 7a). In addition to the Al, Fe, CuAl_2_ phases, a new Al_7_Cu_2_Fe phase is observed at the Al side of IFR-550 (Figure 7b). As the intermediate annealing temperature reaches 600 °C, the main crystalline phases in the peeling surface of the Cu side for IFR-600 are transformed into CuAl_2_, Al, Cu_9_Al_4,_ and FeAl. As to the Al side of IFR-600, the main crystalline phases are CuAl_2_, Fe_3_Al, and Al.

The morphologies of the peeling surface for IFR-450 and the chemical compositions of corresponding crystalline phases are shown in Figure 8 and Table 1. There are two entirely different peeling morphologies on the Cu side and Al side. The ridge-shaped mixture of the Al matrix and Cu-Al IMCs and invaginated Cu matrix can be observed on the Cu side (Figure 8a), which is identified by the EDS results (Table 1). Moreover, many distinct wrinkles are formed at the invaginated Cu matrix, which resulted from the severe shear deformation induced by the plastic difference of the Cu matrix and SUS304 interlayer (Figure 8c). As to the Al side, many SUS304 fragments with uneven sizes are tightly embedded into the Al matrix, which are formed through the random fracture of SUS304 foil in the roll cladding process (Figure 8b). Moreover, these SUS304 fragments’ gaps are filled with a mixture of reticulated Al and Cu-Al IMCs (Figure 8d).

The morphologies of the peeling surface for IFR-550 and the chemical compositions of corresponding crystalline phases are shown in Figure 9 and Table 2. The crystalline phases in the peeling surface of the Cu side for IFR-550 are folded Cu matrix, reticulate Al matrix, and the Cu_9_Al_4_ phase with obvious cracks (Figure 9a), which is consistent with the XRD results. Compared with IFR-450, the content of the Cu_9_Al_4_ phase exhibits a significant increase. The increase in the intermediate annealing temperature leads to the formation of a thick Cu-Al IMCs layer, which is brittle and will be broken in the further rolling process (Figure 9c). SUS304 fragments can still be clearly observed on the peeling surface of the Al side for IFR-550 and their height is smaller than the Cu-Al IMCs (Figure 9b). Furthermore, obvious wrinkles appear on the surface of the SUS304 fragments and a new Al_7_Cu_2_Fe phase is detected on the edge of the SUS304 fragments (Figure 9d). Those above phenomena indicate that the interfacial elemental diffusion has been enhanced by increasing the intermediate annealing temperature.

The morphologies of the peeling surface for IFR-600 and the chemical compositions of corresponding crystalline phases are shown in Figure 10 and Table 3. The main crystalline phases on the peeling surface of the Cu side for IFR-600 are the CuAl_2_ phase with a coarse surface and the vitreous FeAl phase (Figure 10a,c). Compared with IFR-450 and IFR-550, the Cu matrix is undetectable for IFR-600, which implies the transformation of the crack propagation path from the initial Cu/SUS304 interface to the Cu-Al IMCs and Fe-Al IMCs layers. High intermediate annealing temperature results in the formation of an unduly thick IMCs layer and Kirkendall voids, which promotes crack propagation and reduces the interfacial bonding interface [23,24]. Similarly, the SUS304 fragments cannot be observed on the peeling surface of the Al side, only some castle peaks of the CuAl_2_ phase and rock-like Fe_3_Al phase (Figure 10b,d). Furthermore, the reticular Al matrix also disappears. The above results also prove that the interfacial cracks are propagated along the Cu-Al IMCs and Fe-Al IMCs layer and the bonding strength of the Al matrix is higher than these IMCs.

### 3.4. Peeling Strength of the Clad Sheets

Figure 11 shows the peeling strength curves and average peeling strength of the Cu/Al clad sheets with the SUS304 interlayer annealed at different temperatures. To highlight the strengthening effect of the SUS304 interlayer, the peeling curve of the clad sheets without the SUS304 interlayer (SR-0) is also introduced. With the increase in the intermediate annealing temperature, the peeling strength of the clad sheets firstly increase and then decrease (Figure 10a). Compared with IFR-450, the peeling strength of IFR-550 is increased by 11%, from 30.9 N/mm to 34.3 N/mm. Moreover, the fluctuation of the peeling strength curve for IFR-550 is larger than that of IFR-450, indicating the difference in the bonding strength among the SUS304/Cu interface, Cu-Al IMCs layer, and Al matrix being enlarged.

## 4. Discussions

According to the above results in Figure 7, Figure 8, Figure 9 and Figure 10, it can be deduced that the interfacial cracks propagate along the Cu/SUS304 interface, Al matrix, and Cu-Al IMCs layer for IFR-450 and IFR-550. While for IFR-600, the propagation path of interfacial cracks is changed, which is along the Al/SUS304 interface, Al matrix, and Cu-Al IMCs layer. The high intermediate annealing temperature leads to an excessive formation of Al-Fe IMCs and Cu-Al IMCs, resulting in a decrease in the bonding strength for the Al/SUS304 interface and Cu/Al interface. Moreover, Chen et al. [25] indicated that the bonding strength of the Cu/steel interface would be enhanced by increasing the annealing temperature in the range of 400 °C–1000 °C. Thus, the interfacial cracks are more likely formed and prolongated along the Al/SUS304 interface instead of the Cu/SUS304 interface for IFR-600.

The diffusion process across the bonding interface of Cu/Al clad sheets has been verified to conform to the vacancy diffusion [26]. The energy input for interfacial atomic diffusion from two aspects: one is the thermal energy from the intermediate annealing treatment; the other is the shear strain energy provided by the ductility difference between the SUS304 interlayer and Cu/Al matrix, as shown in Figure 12. Our previous research has indicated that increasing the thickness of the SUS304 interlayer can improve the interfacial bonding strength by enlarging the shear strain energy at the bonding interface [15]. In this study, the interfacial atomic diffusion, as well as the interfacial bonding strength, is enhanced by raising the intermediate annealing temperature. While for IFR-600, the peeling strength sharply decreases to 8.3 N/mm, which is even lower than the clad sheets without the SUS304 interlayer, as shown in Figure 11. A high intermediate annealing temperature leads to the formation of unduly thick IMCs layers, which promote the crack propagation and decrease the interfacial bonding strength.

## 5. Conclusions

In this work, the influence of an intermediate annealing temperature on the interfacial microstructure, elemental diffusion, and peeling strength of Cu/Al clad sheet containing a SUS304 interlayer is investigated. The main conclusions are as follows:(1)The interfacial atomic diffusion is significantly enhanced by increasing the intermediate annealing temperature. Nevertheless, after a high-temperature annealing treatment (IFR-600), a liquid phase is formed at the bonding interface and the clear Cu/SUS304/Al interface in IFR-450 and IFR-550 is replaced by the chaotic composite interfaces.(2)For IFR-450 and IFR-550, the interfacial crack is propagated along the Cu/SUS304 interface, Cu-Al IMCs layer, and Al matrix. Compared with IFR-450, the interfacial bonding strength for IFR-550 is improved by 11%, from 30.9 N/mm to 34.3 N/mm, which is proven by the obvious wrinkles on the surface of SUS304 fragments and the formation of a new Al_7_Cu_2_Fe phase.(3)As the intermediate annealing temperature is increased to 600 °C (IFR-600), the propagation path of interfacial crack is changed into the Cu-Al IMCs layer and Fe-Al IMCs layer. The clad sheets broke completely in the unduly thick IMCs layer, resulting in the sharp decrease of interfacial bonding strength, only 8.3 N/mm.

## Figures and Tables

**Figure 1 materials-15-02119-f001:**
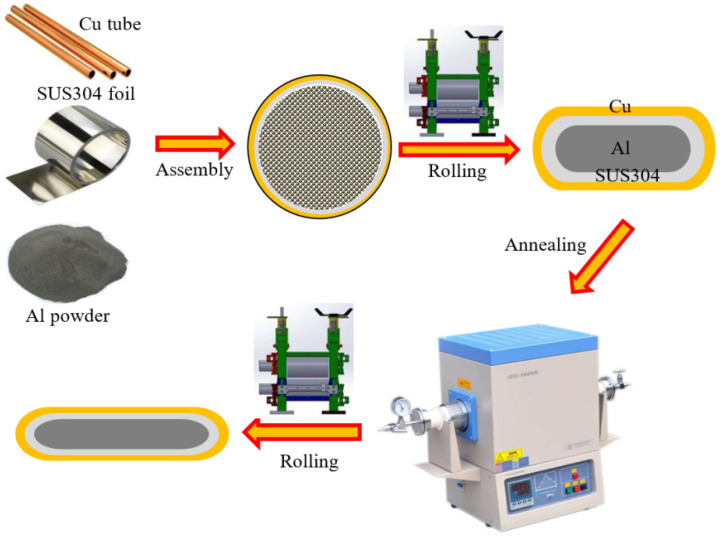
Schematic illustration of the preparation process of Cu/Al clad sheets with SUS304 interlayer.

**Figure 2 materials-15-02119-f002:**
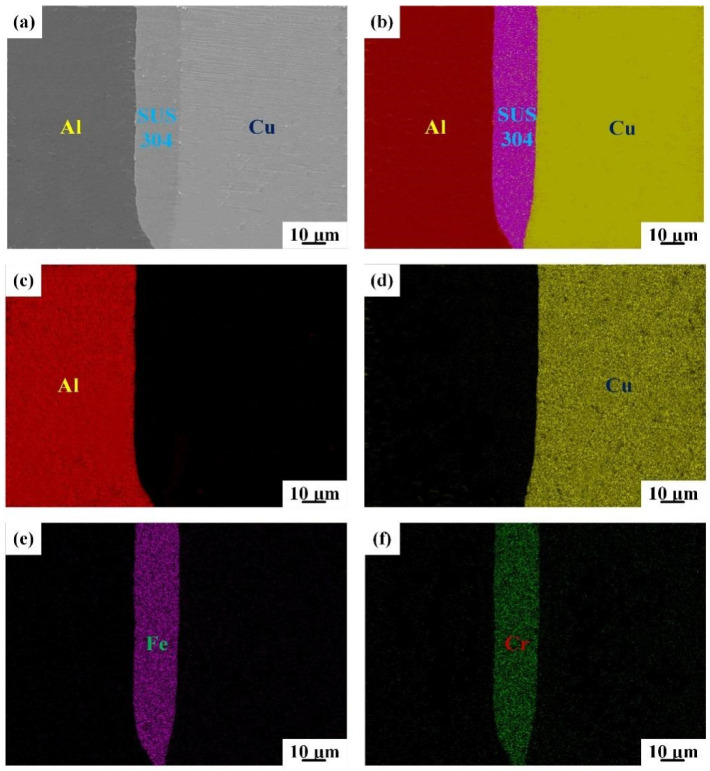
SEM images of the bonding interface for IFR-450 (**a**) and the corresponding EDS mapping: (**b**) EDS layered image, (**c**) Al element, (**d**) Cu element, (**e**) Fe element, (**f**) Cr element.

**Figure 3 materials-15-02119-f003:**
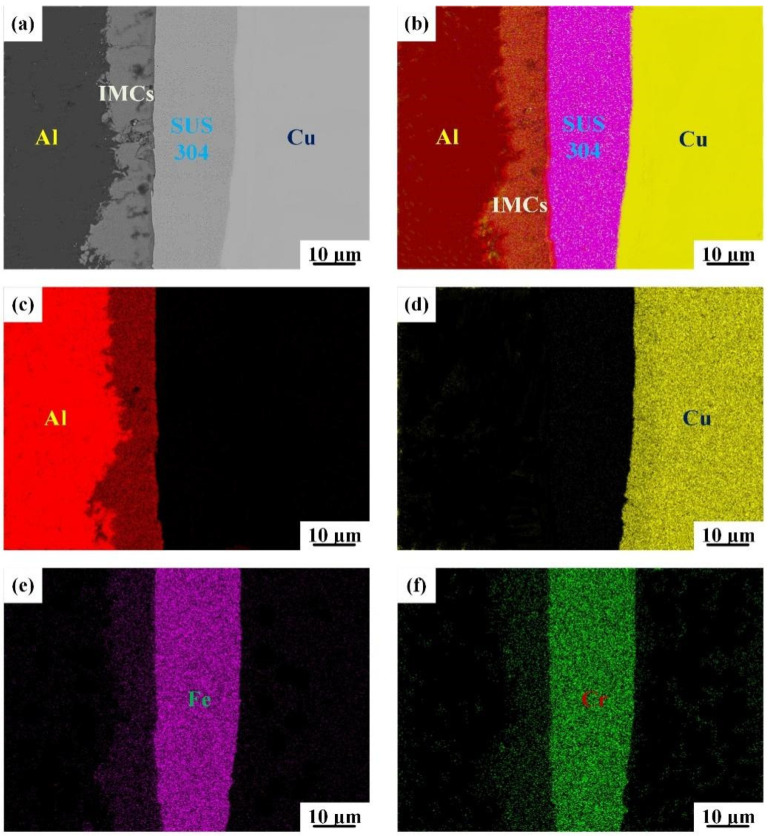
SEM images of the bonding interface for IFR-550 (**a**) and the corresponding EDS mapping: (**b**) EDS layered image, (**c**) Al element, (**d**) Cu element, (**e**) Fe element, (**f**) Cr element.

**Figure 4 materials-15-02119-f004:**
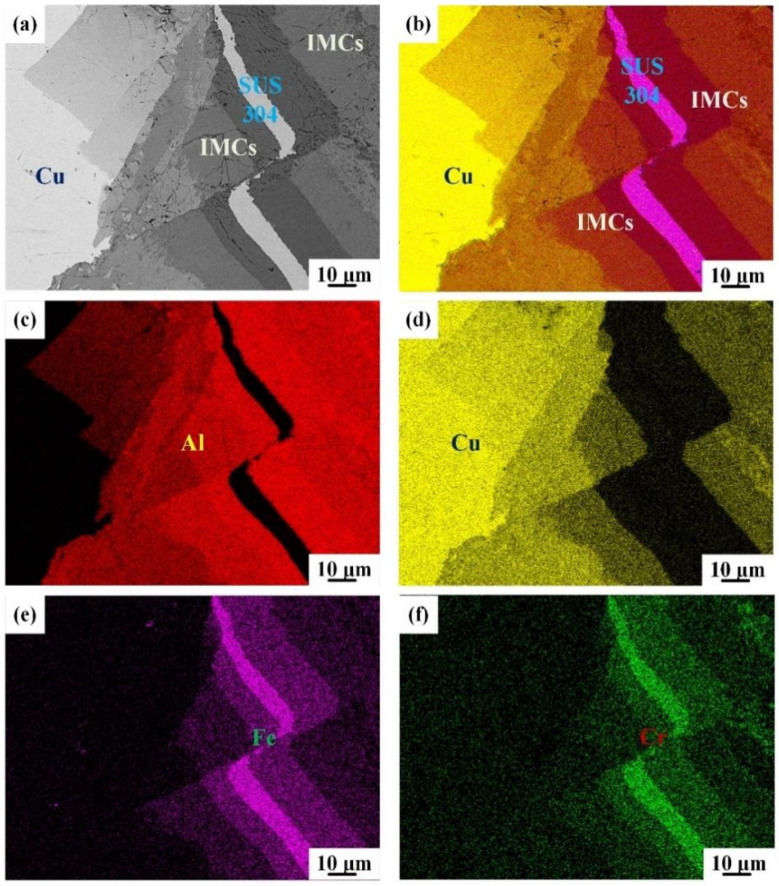
SEM images of bonding interface for IFR-600 (**a**) and the corresponding EDS mapping: (**b**) EDS layered image, (**c**) Al element, (**d**) Cu element, (**e**) Fe element, (**f**) Cr element.

**Figure 5 materials-15-02119-f005:**
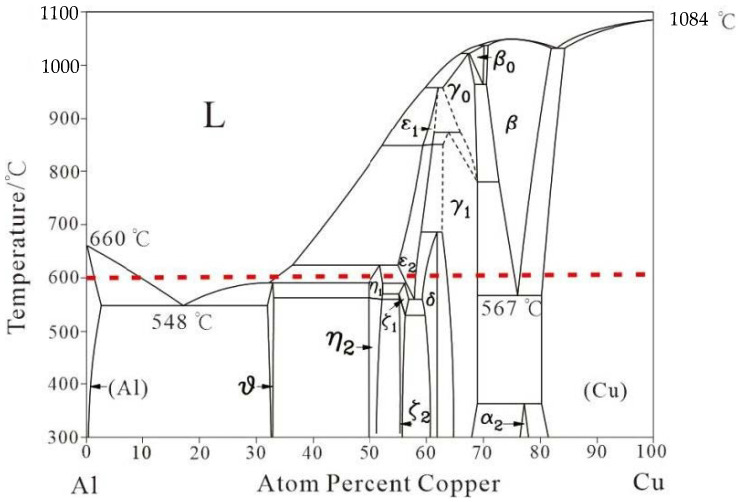
Binary alloy phase diagram of Al-Cu.

**Figure 6 materials-15-02119-f006:**
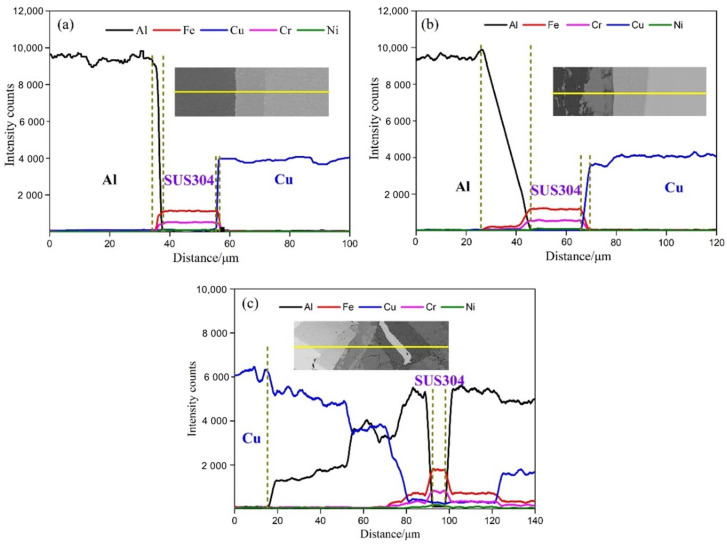
EDS line scan analysis across the Cu/SUS304/Al interface: (**a**) IFR-450, (**b**) IFR-550, (**c**) IFR-600.

**Figure 7 materials-15-02119-f007:**
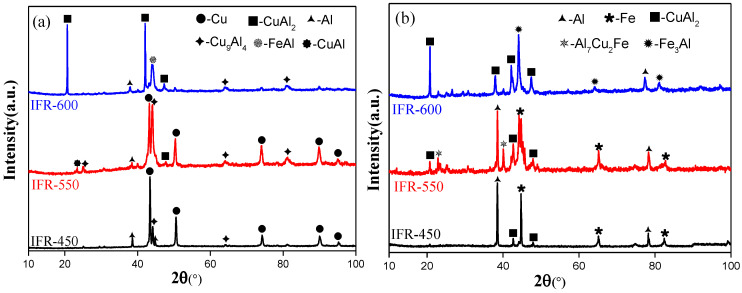
XRD patterns of peeling surface of IFR samples: (**a**) Cu side, (**b**) Al side.

**Figure 8 materials-15-02119-f008:**
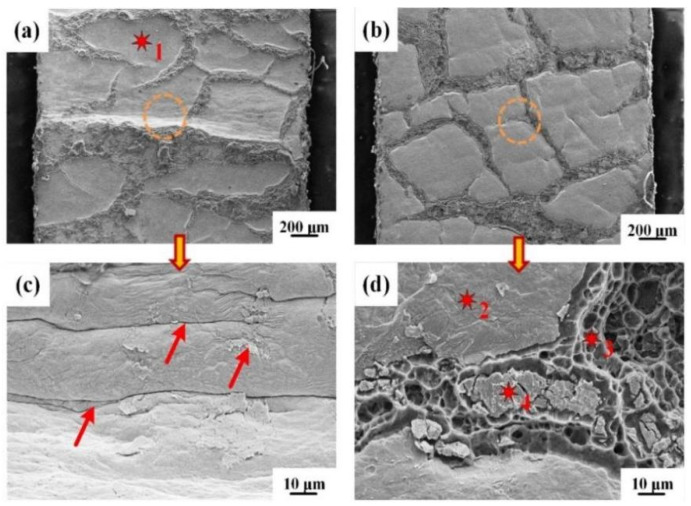
SEM images of Cu and Al surfaces for IFR-450 after peeling test: (**a**) Cu side, (**b**) Al side, (**c**,**d**) enlarged images of the circles in the (**a**,**b**).

**Figure 9 materials-15-02119-f009:**
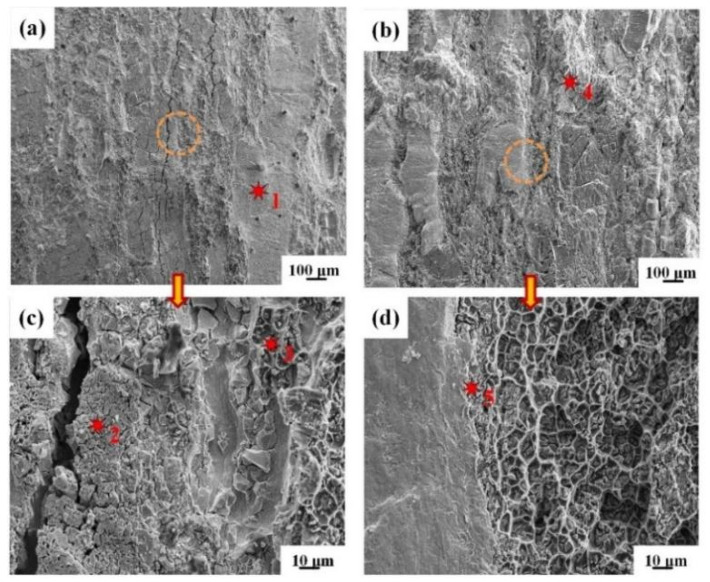
SEM images of Cu and Al surfaces for IFR-550 after peeling test: (**a**) Cu side, (**b**) Al side, (**c**,**d**) enlarged images of the circles in the (**a**,**b**).

**Figure 10 materials-15-02119-f010:**
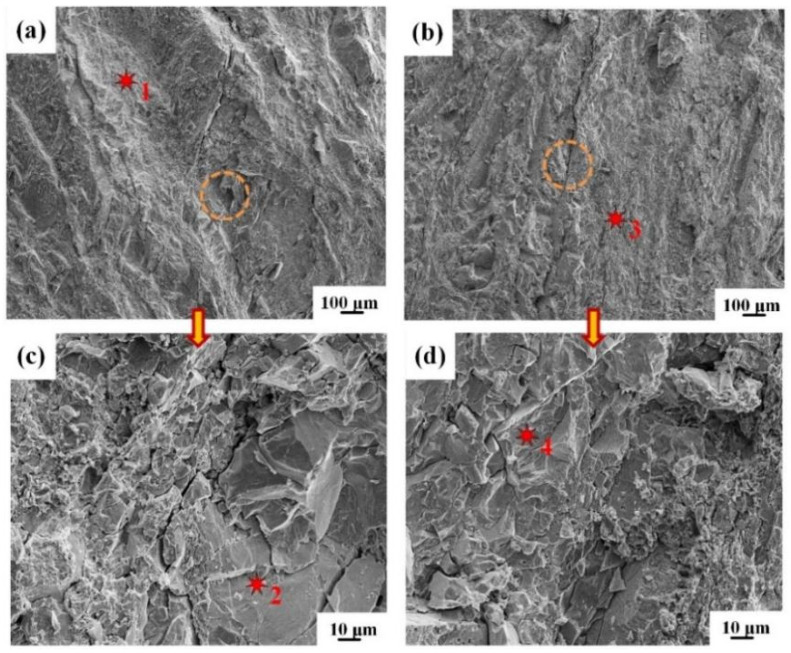
SEM images of Cu and Al surfaces for IFR-600 after peeling test: (**a**) Cu side, (**b**) Al side, (**c**,**d**) enlarged images of the circles in the (**a**,**b**).

**Figure 11 materials-15-02119-f011:**
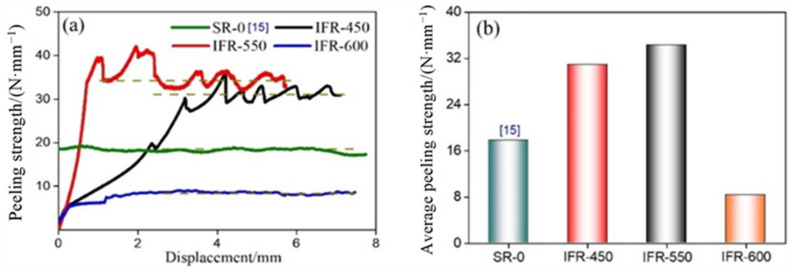
Peeling strength curves (**a**) and average peeling strength (**b**) of Cu/Al clad sheets with SUS304 interlayer annealed at different temperatures.

**Figure 12 materials-15-02119-f012:**
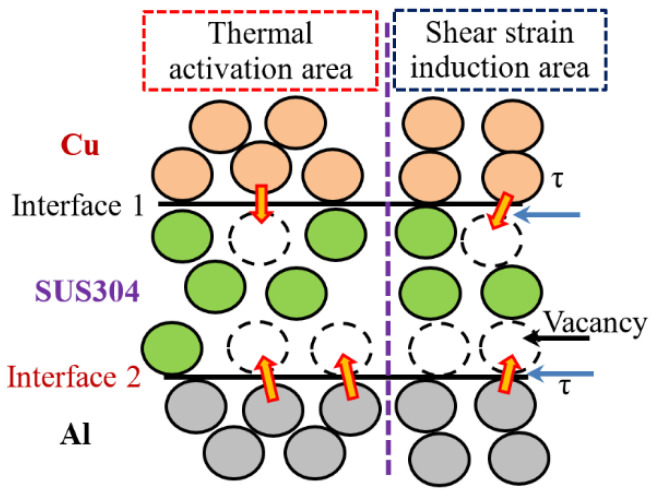
Schematic diagram of vacancy diffusion around the bonding interface.

**Table 1 materials-15-02119-t001:** Chemical composition of corresponding points at the peeling surface of IFR-450 (at. %).

Points	Cu	Al	Fe	Cr	Ni	Phase
1	97.3	2.7	-	-	-	Cu
2	-	-	78.6	16.1	5.3	SUS304
3	1.6	98.4	-	-	-	Al
4	67.2	32.8	-	-	-	Cu_9_Al_4_

**Table 2 materials-15-02119-t002:** Chemical composition of corresponding points at the peeling surface of IFR-550 (at. %).

Points	Cu	Al	Fe	Cr	Ni	Phase
1	92.6	7.4	-	-	-	Cu
2	63.7	36.3	-	-	-	Cu_9_Al_4_
3	3.2	96.8	-	-	-	Al
4	72.5	27.5	-	-	-	Cu_9_Al_4_
5	68.2	19.3	9.1	2.5	0.9	Al_7_Cu_2_Fe

**Table 3 materials-15-02119-t003:** Chemical composition of corresponding points at the peeling surface of IFR-600 (at. %).

Points	Cu	Al	Fe	Cr	Ni	Phase
1	26.8	73.2	-	-	-	CuAl_2_
2	5.2	45.6	43.2	4.3	1.7	FeAl
3	39.5	60.5	-	-	-	CuAl_2_
4	4.3	18.7	59.6	13.8	3.6	Fe_3_Al

## Data Availability

The data presented in this study are available on request from the corresponding author.

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
