# Peer review of "Effect of Annealing Temperature on the Interfacial Microstructure and Bonding Strength of Cu/Al Clad Sheets with a Stainless Steel Interlayer"

_materials, 2022, doi:10.3390/ma15062119_

Round 1

Reviewer 1 Report

Paper: Effect of annealing temperature on the interfacial microstructure and bonding strength of Cu/Al clad sheets with a stainless steel interlayer, 

by Haitao Gao, Hao Gu, Sai Wang, Yanni Xuan and Hailiang Yu

Recommendation: Major revision

General comments:

The subject and object of the work is rather of an appellate nature. However, it can also be interesting scientifically. Unfortunately, the presented work also has errors and the results of the experiments cannot be fully verified in it. Often the explanations are not sufficiently supported by measurement.

Detailed comments:

  1. The abstract does not use abbreviations and explanations, eg IMCs, which are explained for the second time in the 'Introdaction' section. Abbreviations should be removed from the absract.
  2. There was no specific purpose of the research indicated, only what was done was described.
  3. In the 'Materials and Methods' section, it should be explained how the elements were assembled in the first stage of production. Was it a manual connection or is the process automated?
  4. On lines 80-81, too little detail is given regarding the XRD measurements. From the speed and range it is impossible to deduce what the time step and the time counted for this step was. The lack of radiation guests does not allow for the correctness of the phase analysis to be verified. The wavelength, time and angle step is of great importance in the phase analysis because it determines the position of the peaks. Description and other parameters, such as the length of radiation used (tube type), detector, angle step, time step and any filter used, should be added.
  5. Due to the above comment, it is not possible to verify the phase analysis performed in subsection 3.3 and shown in Figure 7. I suspect that the analysis is correct, but not verifiable at this stage. In order to recognize it as correct, complete the information as mentioned above.
  6. The descriptions in figure 2 as well as 3 and 4 are insufficient. The drawings contain 6 images each, and the descriptions seem to indicate only two. It is necessary to add detailed descriptions, preferably in the text, where individual cases are discussed, what the individual images present. Additionally, add a description that allows drawing the presented conclusions.
  7. The expression for diffusion (formula 1) is redundant. Nowhere in the article is it used for calculations. It is sufficient to mention that it has an exponential relationship.
  8. In Figure 4 (a) it can be seen that the phase has been wavy, although the structure is retained. A comment is needed in relation to the production process, especially since the EDS measurements do not directly indicate this nature of the interphase.
  9. The description of the results presented in Figures 9 and 10 is not precise enough, and the explanation of the phenomena taking place is not related to the results of the experiment. These explanations require elaboration and demonstration of clear differences between these cases.
  10. Before the conclusions, there is no general discussion and no correlation between the individual measurements to be shown.
  11. In my opinion, conclusion 1 is not sufficiently explained and supported by material and description in the paper. It is a fact that the interphase is wavy, but it is not known why?

Reviewer 2 Report

The manuscript suits to the journal scope. It is dedicated to one of the important problems of metallurgy, which are interfaces and their modification under temperature. I have some recommendations before publishing.

  • Please, correct font at Figure 5. If you used the figure from some references, do not forger to provide it information in accordance with licence rules.
  • Processing of 304 SS is of interest and a lot of works dedicated to this topic, especially to the SS surface passivation. I suggest to heat up the statement “On the other hand, the interfacial atomic diffusion is also enhanced by the increase of annealing ” by reference: DOI: 10.1016/J.SPMI.2020.106681
  • Please, provide references of card number for XRD data.
  • Could you provide SEM images with lower magnification (higher resolution)?

Reviewer 3 Report

This is a relevant study that falls into the scope of Materials.
It is of high quality - here some minor comments to further improve the quality of this work:

Abstract
Did you measure the peeling strength without annealing? If so please already mention that in the abstract, that would be very interesting.
What Cu (commercially pure) and particularly what Al did you use, can be quickly named in the abstract, otherwise I suspect both were in pure form. I would also mention the thickness of the materials already in the abstract and the different temperatures you examined. This is usually very helpful to readers.

line 33-34: Diffusion bonding, TLP bonding and diffusion brazing (https://doi.org/10.3390/met10050613) are other importent processes to produce Cu?SUS304/Al clad sheets or with only two of these materials. Should be quickly added to your list.
Line 40 : why does if have a significant influence?

Maybe provide the underlying diffusion equation - not really needed, I saw this in a couple of other publications. OK I see you have that later on in Eq. 1 good
So you use pure Al I guess? mention that. Fig. 5 is really useful here.

Maybe compine Fig. 8 and Table 1 - so that the points and their chemical composition are in one Figure/on one page
Same for Fig. 9 and Table 2
Same for Fig. 10 and Table 3

Good - well done

Round 2

Reviewer 3 Report

All comments have been addressed well - this manuscript can be accepted for publication now.